# Evaluation of the Diagnostic Accuracy of a New Biosensors-Based Rapid Diagnostic Test for the Point-Of-Care Diagnosis of Previous and Recent Dengue Infections in Malaysia

**DOI:** 10.3390/bios11050129

**Published:** 2021-04-22

**Authors:** Zhuo Lin Chong, Hui Jen Soe, Amni Adilah Ismail, Tooba Mahboob, Samudi Chandramathi, Shamala Devi Sekaran

**Affiliations:** 1Centre for Communicable Diseases Research, Institute for Public Health, National Institutes of Health, Ministry of Health, Persiaran Setia Murni, Setia Alam, Shah Alam 40170, Selangor, Malaysia; 2Department of Medical Microbiology, Faculty of Medicine, University of Malaya, Jalan Profesor Diraja Ungku Aziz, Kuala Lumpur 50603, Malaysia; jeanies215@gmail.com (H.J.S.); amniadilah18@gmail.com (A.A.I.); tooba666@hotmail.com (T.M.); chandramathi@um.edu.my (S.C.); 3Faculty of Medical & Health Sciences, UCSI University, Jalan Menara Gading, Cheras, Kuala Lumpur 56000, Malaysia

**Keywords:** dengue, biosensors, immuno-magnetic agglutination assay, point-of-care diagnosis, rapid diagnostic test, previous and recent dengue, diagnostic accuracy, seroepidemiological survey, vaccination strategy

## Abstract

Dengue is a major threat to public health globally. While point-of-care diagnosis of acute/recent dengue is available to reduce its mortality, a lack of rapid and accurate testing for the detection of previous dengue remains a hurdle in expanding dengue seroepidemiological surveys to inform its prevention, especially vaccination, to reduce dengue morbidity. This study evaluated ViroTrack Dengue Serostate, a biosensors-based semi-quantitative anti-dengue IgG (immunoglobulin G) immuno-magnetic agglutination assay for the diagnosis of previous and recent dengue in a single test. Blood samples were obtained from 484 healthy participants recruited randomly from two communities in Petaling district, Selangor, Malaysia. The reference tests were Panbio Dengue IgG indirect and capture enzyme-linked immunosorbent assays, in-house hemagglutination inhibition assay, and focus reduction neutralization test. Dengue Serostate had a sensitivity and specificity of 91.1% (95%CI 87.8–93.8) and 91.1% (95%CI 83.8–95.8) for the diagnosis of previous dengue, and 90.2% (95%CI 76.9–97.3) and 93.2% (95%CI 90.5–95.4) for the diagnosis of recent dengue, respectively. Its positive predictive value of 97.5% (95%CI 95.3–98.8) would prevent most dengue-naïve individuals from being vaccinated. ViroTrack Dengue Serostate’s good point-of-care diagnostic accuracy can ease the conduct of dengue serosurveys to inform dengue vaccination strategy and facilitate pre-vaccination screening to ensure safety.

## 1. Introduction

Dengue is a mosquito-borne infectious disease caused by the dengue virus (DENV) with four serotypes (DENV-1 to DENV-4) from the genus Flavivirus [1,2]. It has become a public health threat affecting more than 100 countries globally [3,4,5]. Every year, approximately 390 million dengue infections occur worldwide, most of which happen in tropical and subtropical regions [2,6]. The Global Burden of Disease Study 2013 estimated that in that year alone, dengue illness cost the world US$ 8.9 billion to treat [4], and was responsible for 566,000 years lived with disability and 576,900 years of life lost [7].

While dengue deaths can be prevented with the early detection and appropriate treatment of dengue cases, the prevention and control of dengue transmission requires a more coordinated effort in disease surveillance and integrated vector management [1,2]. In addition, due to the presence of asymptomatic infection and its nonspecific clinical manifestations, more accurate estimation of the true burden of dengue is required to inform the effectiveness of current preventive measures, as well as the future implementation of a dengue vaccination program [2,8,9].

The estimation of the true burden of dengue can only be achieved with the help of diagnostic tests; in particular, serology tests for the detection of anti-dengue antibodies [10,11,12]. The development of laboratory diagnostics in the past decades has made this possible, but they are costly and time-consuming [13]. The arrival of rapid diagnostic tests (RDTs) has made point-of-care diagnosis possible, but all dengue RDTs developed to date, even serology tests, catered only for the diagnosis of acute or recent dengue infection [13,14,15,16,17]. These tests are mainly paper-based tests with a fixed detection threshold, making their serology components unsuitable to determine the status of previous dengue infection due mainly to decreasing IgG (immunoglobulin G) levels post-infection below their detection threshold [14,18], although they are able to diagnose recent dengue. In addition, RDTs are not as accurate as laboratory-based diagnostics [14,15,16].

The breakthrough in the development of a biosensors-based dengue RDT with quantitative read-out and objective interpretation makes it possible to strike a balance between rapidity and accuracy of dengue diagnosis [19,20]. A rapid yet accurate biosensors-based dengue serology RDT capable of detecting not just recent but also previous dengue infection has the potential to improve dengue surveillance through seroepidemiological survey and inform dengue vaccination strategies [9,11,12,21]. This article aimed to evaluate the diagnostic accuracy of a new biosensors-based RDT for the diagnosis of previous and recent dengue infections.

## 2. Materials and Methods

### 2.1. Study Design

This is a prospective cross-sectional diagnostic test accuracy evaluation study. Publication of this study complies with the highest requirement in the Standards for the Reporting of Diagnostic Accuracy Studies (STARD) guidelines [22].

### 2.2. Participants

The study was carried out in a highly urbanized district called Petaling in the state of Selangor, where dengue incidence was among the highest in Malaysia [23]. Two communities were selected randomly from a list of all communities covered by the District Health Office for dengue vector control activities, with one community each from those with the highest and the lowest reported dengue cases from the years 2013 to 2017. A total of 500 participants were required based on the single proportion sample size formula to achieve <10% margin of error for the expected dengue seroprevalence in these communities with type I error of 5% [24]. These participants were recruited from 250 households selected randomly from both communities proportionate to their population size, with an expectation of 4 people in each household and a 50% overall attrition rate [25]. All residents aged ≥ nine months who lived in the selected household address in the past six months were included in the study. People with current febrile illness, history of other flaviviral infection or vaccination, or risk factors of complications from blood taking were excluded.

The recruitment of participants lasted from 18 August to 26 October 2018. All consented participants were interviewed by trained field researchers using a structured questionnaire to capture socio-demographic characteristics. Both capillary and venous blood specimens were sampled from each participant. Capillary blood was meant for the evaluated index test that was conducted on-site. It was drawn from the prick site using a capillary tube and immediately transferred to a vial containing buffer solution. Venous blood specimens were collected in plain tubes, chilled, and transferred at the end of each workday to a virology laboratory in the University of Malaya, Kuala Lumpur, Malaysia, where they were centrifuged, aliquoted, and stored at −80 °C until reference tests were conducted in batches.

### 2.3. Index Test

ViroTrack Dengue Serostate (BluSense Diagnostics, Copenhagen, Denmark) was the evaluated index test. It is a biosensor composed of a polymer centrifugal microfluidic cartridge embedded with dry reagent and a portable opto-magnetic reader—BluBox (Figure 1) [26]. It comes with a vial of buffer solution used to dilute the test sample (blood/plasma/serum). The reader is CE marked and in use for other tests. The ViroTrack Dengue Serostate is a semi-quantitative anti-dengue immunoglobulin G (IgG) assay based on the immuno-magnetic agglutination (IMA) principle. The scientific principle of this assay and the design of the microfluidic are described in detail elsewhere [19,27,28,29]. Briefly here, 10 mcl capillary blood is diluted in the buffer solution. After mixing well, 10 mcl of the diluted sample is pipetted into the loading chamber of the cartridge. The cartridge is then inserted into the BluBox, in which the sample is centrifuged, metered, and mixed with nanoparticles (MNPs) coated with DENV antigen. This antigen captures any anti-dengue IgG present in the test sample and forms sandwich agglutination after magnetic incubation. Using an oscillating magnetic field, the MNPs are forced to rotate, which then modulate the intensity of a passing laser beam. The difference between the modulated light wave and the applied field is directly proportional to the concentration of anti-dengue IgG in the test sample. It is measured by a photodetector with Blu-ray optical pickup and given in a relative unit (BluSense IMA unit), which the BluBox interprets according to set thresholds, i.e., presence of previous dengue infection if ≥8 units and recent dengue infection if ≥140 units [20]. The duration from cartridge insertion to result is around 8 min. Index test was run on-site by trained field researchers immediately after capillary blood collection. As such, they were blinded to the participants’ previous and recent dengue status as defined by the reference tests below.

### 2.4. Reference Standard

The reference standard for previous dengue diagnosis comprised of three different reference tests for the detection of anti-dengue antibodies, namely (1) commercially available Panbio Dengue IgG indirect enzyme-linked immunosorbent assay (ELISA) (Abbott, Chicago, IL, USA), (2) in-house hemagglutination inhibition (HI) assay, and (3) in-house focus reduction neutralization test (FRNT) adapted from the plaque reduction neutralization test (PRNT). HI remains the gold standard for the titration and subtyping of antibodies due to its high accuracy and its ability to determine the exact titer of virus-neutralizing antibodies [30]. Meanwhile, the serotype-specific PRNT/FRNT is considered the gold standard for the differential serodiagnosis of dengue virus [31]. However, there is a chance that HI might not detect the antibodies against other viral components aside from the hemagglutinating protein, despite being cross-reactive with other flaviviral infections. Similarly, a certain degree of cross-reactivity exists between the four serotypes in PRNT/FRNT. As such, IgG ELISA with sensitivity that is comparable to HI, but significantly more specific than HI, was also selected to be part of the composite reference standard [32].

Previous dengue diagnosis was defined as positive when a sample was tested negative to only one out of three reference tests above, which were in turn defined as follows: (1) for IgG indirect ELISA (<9, 9–11, and >11 Panbio units were positive, equivocal, and negative, respectively), (2) for HI (titer of <1:10 and ≥1:10 were negative and positive, respectively). For FRNT, the titer reported was reciprocal of the highest serum dilution capable of 95% plaque reduction compared to the serum-free control (FRNT_95_). Titer of <1:10 and ≥1:10 for FRNT_95_ against a specific DENV serotype were defined as negative and positive tests to that serotype, respectively. Finally, FRNT_95_ positive was defined as a positive test to any of the individual serotype-specific FRNT_95_.

On the other hand, recent dengue status was defined as positive when a sample was tested positive to Panbio Dengue IgG capture ELISA (Abbott, Chicago, IL, USA), where <18, 18–22, and >22 Panbio units were positive, equivocal, and negative, respectively.

All serum specimens extracted from the venous blood samples of all participants were tested on all the above reference tests from 6 December 2018 to 19 April 2019 by trained laboratory researchers blinded to the results of the index test. Commercial tests were conducted and interpreted according to the manufacturer’s instructions [33,34,35], while in-house tests were performed and interpreted as described previously [30,31,32,36,37,38,39,40,41]. (Appendix A).

### 2.5. Analysis

Descriptive analysis was used to describe socio-demographic characteristics of the participants and the dengue status of their samples according to the reference standards and by individual reference tests. Diagnostic accuracy and their 95% confidence intervals (95%CI) were derived from 2 × 2 tables comparing the results of the index test and the dengue status as defined by the reference standards and by individual reference tests, where samples were divided into true positives (TP), false positives (FP), true negatives (TN), and false negatives (FN). Accuracy estimates computed were (1) sensitivity (SN) = TP/(TP + FN); (2) specificity (SP) = TN/(TN + FP); (3) positive predictive value (PPV) = TP/(TP + FP); and (4) negative predictive value (NPV) = TN/(TN + FN) [42]. Finally, the correlation of ViroTrack relative unit and the Panbio units for both indirect and capture ELISAs was visualized using scatterplot and tested using Spearman’s rho (ρ), which was defined as very strong (≥0.8), moderately strong (0.6–0.8), fair (0.3–0.5), and poor (<0.3) [43]. A p of <0.05 was accepted as statistical significance of this correlation. All statistical analyses were performed using STATA version 12 (StataCorp, College Station, TX, USA). Any sample with inconclusive or missing results of any test was excluded from the final analysis.

### 2.6. Ethical Statement

This study adhered to the revised 2013 Declaration of Helsinki [44]. Ethics approval was obtained from the Medical Research and Ethics Committee, Ministry of Health Malaysia (NMRR-17-853-34393) and the University Malaya Medical Center Medical Research Ethics Committee (MRECID.NO: 2017426-5171). All participants provided written informed consent and, where applicable, minor assent.

## 3. Results

### 3.1. Description of the Study Participants

Out of the 533 eligible participants identified, 515 (96.6%) consented to participate in this study, and the results for 484 (90.8%) were included in the final analysis. Among the latter, 358 tested as positive on ViroTrack and 126 were negative. Reference standard was available for all of them (Figure 2).

The age of these 484 participants ranged from 1.0 to 83.7 years, and averaged at 32.0 (s.d. 18.4) years. There were 250 (51.7%) male participants. The majority were of Malay ethnicity (373, 77.1%), followed by Indian descent (52, 10.7%,) and Chinese descent (31, 6.4%), while 8 (1.7%) people were natives of East Malaysia and 20 (4.1%) were foreigners (data not shown).

Out of the 484 participants, 363 (75.0%) tested positive with IgG indirect ELISA and 356 (73.6%) with HI. Around 54.8–61.6% of participants tested positive with FRNT_95_ against a specific DENV serotype, which made up to 414 (85.5%) positive tests against any DENV serotype on FRNT_95_. Based on the abovementioned reference standards, a total of 383 (79.1%) participants were infected with dengue previously, while 41 (8.5%) were infected with dengue recently (Table 1).

### 3.2. Diagnostic Accuracy of ViroTrack Dengue Serostate

ViroTrack Dengue Serostate’s SN was 91.1% (95%CI 87.8–93.8), and its SP was 91.1% (95%CI 83.8–95.8) for the diagnosis of previous dengue infection. The PPV and NPV were 97.5% (95%CI 95.3–98.8) and 73.0% (95%CI 64.4–80.5), respectively (Table 2).

For the diagnosis of recent dengue, the SN and SP of ViroTrack Dengue Serostate were 90.2% (95%CI 76.9–97.3) and 93.2% (95%CI 90.5–95.4), respectively. Its PPV was 55.2% (95%CI 42.6–67.4), while its NPV was 99.0% (95%CI 97.6–99.7) (Table 2).

When compared to the individual reference tests used to construct the reference standard for previous dengue, ViroTrack accuracy varied (Table 3). When IgG indirect ELISA alone was used as a standard, ViroTrack had 95.3% (95%CI 92.6–97.3) SN, 90.1% (95%CI 83.3–94.8) SP, 96.7% (95%CI 94.2–98.3) PPV, and 86.5% (95%CI 79.3–91.9) NPV. When compared to HI only, the SN, SP, PPV, and NPV of ViroTrack were 91.6% (95%CI 88.2–94.2), 75.0% (95%CI 66.6–82.2), 91.1% (95%CI 87.6–93.8), and 76.2% (95%CI 67.8–83.3), respectively. Lastly, its SN and SP when compared to FRNT_95_ alone were 75.6% (95%CI 71.2–79.7) and 35.7% (95%CI 24.6–48.1), respectively; while the PPV and NPV were 87.4% (95%CI 83.5–90.7) and 19.8% (95%CI 13.3–27.9), respectively.

### 3.3. Correlation of ViroTrack Dengue Serostate and Panbio Dengue IgG ELISA

The correlation between ViroTrack Dengue Serostate and both Panbio Dengue IgG indirect and capture ELISAs is visualized in Figure 3. The BluSense IMA unit was positively correlated to the Panbio unit for both ELISAs, where the curve appeared to be logarithmic for indirect ELISA and exponential for capture ELISA. The Spearman’s ρ for the BluSense IMA unit and the Panbio unit was 0.8728 (*p* < 0.0001) for indirect ELISA and 0.9238 (*p* < 0.0001) for capture ELISA, indicating a very strong correlation between ViroTrack Dengue Serostate and both ELISAs that was statistically significant.

## 4. Discussion

This study found that ViroTrack Dengue Serostate performed well for the diagnosis of previous and recent dengue infections when compared to the reference standards, with SN and SP for both diagnoses above 90%. The PPV for the diagnosis of previous dengue and the NPV for the diagnosis of recent dengue were close to 100% (Table 2). When compared to each individual laboratory test used in the definition of the reference standard for previous dengue, its accuracy point estimates were above 75% for all tests, except for FRNT_95_ (Table 2).

The main reason for ViroTrack’s low SP and, correspondingly, PPV, as compared to FRNT_95_, was due to the cross-reactivity of our FRNT [10,38]. PRNT/FRNT is performed in-house with varied assay methodology. Its titers can be affected by many factors such as difference in cell line, viral passage, etc. [38,45]. In our case, while around 60% of the samples had a titer of ≥1:10 for FRNT_95_ against a specific DENV serotype, as many as 85.5% tested positive for any one of the serotypes, much higher than that of HI and IgG indirect ELISA. The FRNT positive rate would be even higher at 93.2% for any serotype (data not shown) if the cut-off of 90% plaque reduction were used. With a higher positive rate, it was possible that our FRNT_95_ misclassified some negative tests as positive. Although ViroTrack was able to correctly classify these samples as negative, these supposedly TN results were deemed as FN in the 2 × 2 table. The increase in FN and decrease in TN then falsely decreased the SN, SP, and NPV of ViroTrack when compared to FRNT_95_ alone [46]. This decrease was more notable for NPV as TN was used as the denominator, while FN was the numerator in its calculation. However, PPV stayed totally unaffected and reflected the true value.

While FRNT_95_ alone was an imperfect gold standard, IgG ELISA and HI also have their fair share of cross-reactivity when used alone [1]. We overcame this shortcoming with the use of a composite reference standard [46]. By defining previous dengue positive as those tested negative to only one out of three different reference tests, we reduced the number of falsely classified dengue-naïve cases and increased the specificity of the reference standard. Here, our reference standard defined 79.1% of our participants as previously infected with DENV, which was in line with the seropositive rates found among urban dwellers in Malaysia [24,47]. As such, our reference standard provided an accurate and realistic representation of the dengue immunological profile of our participants.

ViroTrack has a huge potential for application. First of all, paper-based dengue serology RDTs with a fixed positivity threshold are built exclusively for the diagnosis of acute/recent dengue, as it is not profitable for the manufacturers to design another one specifically for serological surveillance [13,14,15,16,17,48]. On the other hand, conducting dengue seroprevalence study using laboratory-based serology tests has higher specimen requirements and is logistically tedious [49]. ViroTrack, which is semi-quantitative, can fill this gap in disease surveillance, as the same IgG test can produce one numerical output that can be interpreted according to different positivity thresholds for the diagnosis of previous and recent dengue infections, effectively reducing its development costs. Secondly, its combination with previously validated ViroTrack Dengue Acute would be helpful not only in increasing the latter’s SN in diagnosing acute/recent dengue, but would also concurrently provide ongoing dengue seroprevalence data [20].

Finally, and more importantly, ViroTrack Dengue Serostate can be used as a component of a dengue vaccination strategy. Currently, the only licensed dengue vaccine, Dengvaxia, was developed by Sanofi Pasteur. Although moderately efficacious for seropositive individuals, this vaccine put dengue-naïve subjects at a greater risk of hospitalization for severe dengue due to its lower efficacy among them [8,9]. As such, an individual pre-vaccination screening was recommended prior to the use of Dengvaxia. Otherwise, vaccination without individual screening can also be considered in places where dengue seroprevalence is ≥80% among the population below 9 years old [9]. The application of ViroTrack in the latter case was as discussed above. For individual pre-vaccination screening, ViroTrack Dengue Serostate is user-friendly, portable, and rapid, making it useful to be administered at point-of-care immediately before vaccination. However, more importantly, its high PPV in determining previous dengue status means very few dengue-naïve children will be vaccinated, and therefore even fewer will be hospitalized for severe dengue in the future.

To the best of our knowledge, this is the first study that evaluates the diagnostic accuracy of a biosensors-based RDT, ViroTrack, for the diagnosis of previous and recent dengue. As mentioned above, all the commercial dengue RDTs available currently were intended for the diagnosis of acute/recent dengue. As such, the majority of the evaluation studies conducted on them were solely for that purpose. To date, only two discontinued paper-based dengue IgG RDTs were evaluated for the diagnosis of previous dengue. Their SP was similar at 83.3%, but their SN were only 43.2% and 6.8% [48]. For acute/recent dengue diagnosis, the accuracy of paper-based dengue IgG RDT varied from 38.8 to 90.1% for SN, and 92.5–100.0% for SP [20]. However, these results are not directly comparable to the results of our study due to heterogeneity in the study design and sample population. Further studies that evaluate different RDTs head-to-head are required for direct comparison [20].

The strength of our study lies in its sound methodology. It was conducted among community dwellers in an urban setting in Malaysia using a cross-sectional prospective design and a random sampling. Coupled with a good response rate, the selection bias in this study was negligible. Measurement bias was eliminated by the blinding of the performers of index tests and reference tests, and by ensuring all participants received the same index test and reference tests [46,50]. Finally, this study was reported according to the STARD guidelines for greater transparency and accountability [22]. The limitation of our study was mainly the cross-reactivity of FRNT_95_. However, the bias that arose from the imperfect gold standard was rectified with the use of a composite reference standard. Lastly, as this article was the first evaluation study published for ViroTrack Dengue Serostate, although its diagnostic accuracy for the diagnosis of previous and recent dengue infection is applicable to our population, it might differ in other populations. More evaluation studies are recommended among different populations prior to its application on them, especially as a pre-vaccination screening tool.

## 5. Conclusions

ViroTrack Dengue Serostate, a semi-quantitative biosensors-based dengue IgG RDT, was accurate in diagnosing previous and recent dengue infections in Malaysia with a single test. Its potential applications are (1) community-based dengue seroepidemiological surveillance to estimate the burden of dengue to guide dengue prevention and control measures, including vaccination programs; (2) individual pre-vaccination screening immediately prior to dengue vaccination; and (3) combination with its previously evaluated sibling to improve the accuracy in diagnosing acute/recent dengue. Further evaluation among the intended populations is recommended prior to any of its applications above.

## Figures and Tables

**Figure 1 biosensors-11-00129-f001:**
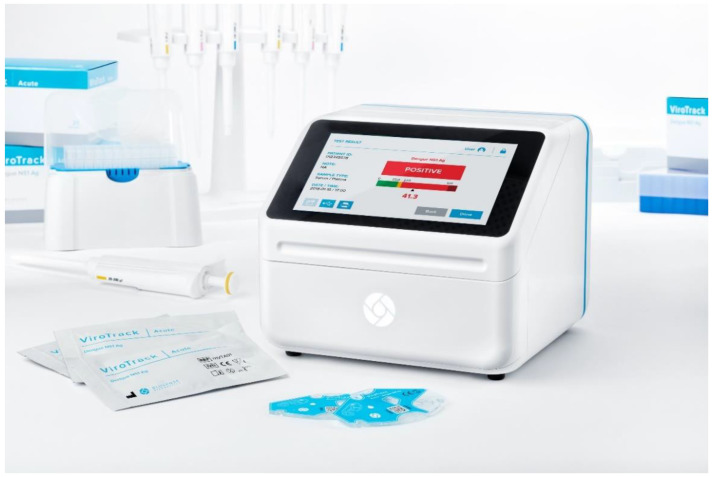
ViroTrack and BluBox.

**Figure 2 biosensors-11-00129-f002:**
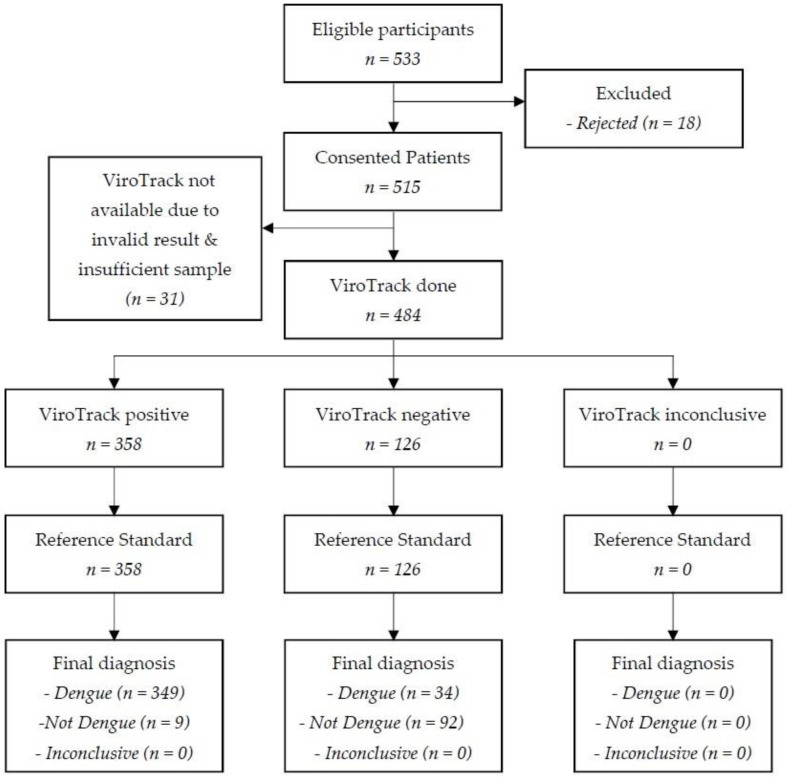
STARD flow diagram for the evaluation of ViroTrack Dengue Serostate.

**Figure 3 biosensors-11-00129-f003:**
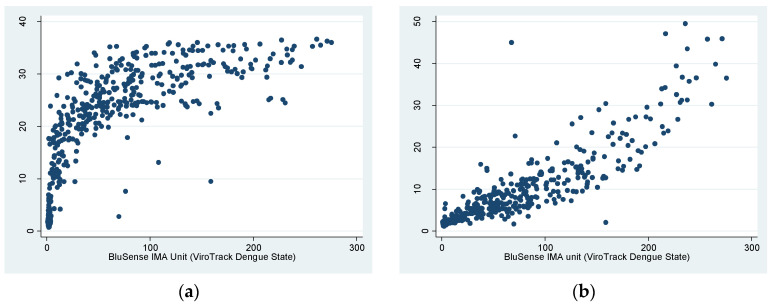
Correlation between the ViroTrack Dengue Serostate’s BluSense IMA Unit and Panbio Unit of (**a**) Panbio Dengue IgG indirect ELISA and (**b**) Panbio Dengue IgG capture ELISA.

**Table 1 biosensors-11-00129-t001:** Characterization of the 484 study participants.

Sample Characteristics	n (%)		
	Positive	Equivocal	Negative
(I) Previous dengue status * (a) IgG indirect ELISA only (b) Hemagglutination inhibition only (c) FRNT_95_ only (i) FRNT_95_ against DENV-1 only (ii) FRNT_95_ against DENV-2 only (iii) FRNT_95_ against DENV-3 only (iv) FRNT_95_ against DENV-4 only	383 (79.1)	-	101 (20.9)
	363 (75.0)	11 (2.3)	110 (22.7)
	356 (73.6)	-	128 (26.4)
	414 (85.5)	-	70 (14.5)
	298 (61.6)	-	186 (38.4)
	299 (61.8)	-	185 (38.2)
	296 (61.2)	-	188 (38.8)
	265 (54.8)	-	219 (45.2)
(II) Recent dengue status **	41 (8.47)	12 (2.48)	431 (89.05)
(IgG capture ELISA only)			

* Previous dengue status was defined as positive when a sample tested negative to only one out of three reference tests, which were in turn defined as follows: (a) for IgG indirect ELISA (<9, 9–11, and >11 Panbio units were positive, equivocal, and negative, respectively), (b) for hemagglutination inhibition (titer of <1:10 and ≥1:10 were negative and positive, respectively). For FRNT, the titer reported was reciprocal of the highest serum dilution capable of 95% plaque reduction compared to serum-free control (FRNT95). Titer of <1:10 and ≥1:10 for FRNT95 against a specific DENV serotype were defined as negative and positive tests for that particular serotype, respectively. Finally, a positive test to FRNT95 was defined as a positive test to any of the individual serotype-specific FRNT95. ** Recent dengue status was defined as positive when a sample tested positive to Panbio Dengue IgG capture ELISA, where <18, 18–22, and >22 Panbio units were positive, equivocal, and negative, respectively).

**Table 2 biosensors-11-00129-t002:** Accuracy of ViroTrack Dengue Serostate for the Diagnosis of Previous and Recent Dengue.

Estimate, % (95%CI)	Previous Dengue	Recent Dengue
Sensitivity	*349/383* 91.1 (87.8–93.8)	*37/41* 90.2 (76.9–97.3)
Specificity	*92/101* 91.1 (83.8–95.8)	*413/443* 93.2 (90.5–95.4)
Positive Predictive Value	*349/358* 97.5 (95.3–98.8)	*37/67* 55.2 (42.6–67.4)
Negative Predictive Value	*92/126* 73.0 (64.4–80.5)	*413/417* 99.0 (97.6–99.7)

Note: The italic numbers before the estimates and their 95%CI are the numbers used to derive them.

**Table 3 biosensors-11-00129-t003:** Accuracy of ViroTrack Dengue Serostate as compared to individual reference tests used in the reference standard of previous dengue diagnosis.

Estimate, % (95%CI)	IgG Indirect ELISA Only	Hemagglutination Inhibition Only	FRNT_95_ Only
Sensitivity	*346/363* 95.3 (92.6–97.3)	*326/356* 91.6 (88.2–94.2)	*313/414* 75.6 (71.2–79.7)
Specificity	*109/121* 90.1 (83.3–94.8)	*96/128* 75.0 (66.6–82.2)	*25/70* 35.7 (24.6–48.1)
Positive Predictive Value	*346/358* 96.7 (94.2–98.3)	*326/358* 91.1 (87.6–93.8)	*313/358* 87.4 (83.5–90.7)
Negative Predictive Value	*109/126* 86.5 (79.3–91.9)	*96/126* 76.2 (67.8–83.3)	*25/126* 19.8 (13.3–27.9)

Note: The italic numbers before the estimates and their 95%CI are the numbers used to derive them.

## Data Availability

The data presented in this study are available on request from the corresponding author. The data are not publicly available due to agreement of confidentiality.

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
