# Peer review of "Evaluation of the Diagnostic Accuracy of a New Biosensors-Based Rapid Diagnostic Test for the Point-Of-Care Diagnosis of Previous and Recent Dengue Infections in Malaysia"

_biosensors, 2021, doi:10.3390/bios11050129_

Round 1

Reviewer 1 Report

This work describes a biosensor-based semi-quantitative anti-dengue IgG immunomagnetic agglutination assay. This sensor allows the diagnosis of past and recent dengue fever in a single test. This sensor works well, but the whole work is not very innovative. However, this work is well suited for Biosensors,and I think it can be received after the major revision.

  1. The authors put how to assemble the biosensor in the supporting material, and I suggest that the authors could have put some of the important parts of it in the main text. Regarding the design of the biosensor, I suggest that the authors put a schematic diagram.
  2. Line 267: In fact, no laboratory test is perfect. Please avoid using such language for academic writing.
  3. In this work, the authors carried out meaningful work, but did not focus on the design and optimization of the biosensor. Do you already have a previous publication on this biosensor?

Overall, this work is acceptable for publication, but the authors will also need to provide answers to these questions above.

Reviewer 2 Report

It would be good to indicate in lines 55-57 why the RDTs are not suitable for the detection of previous Dengue. Is this because the serological levels are decreasing too much and the sensitivity of the test is not good enough compared to the levels for acute or recent Dengue?

To improve the understanding of the reader, a schematic of the ViroTrack assay will greatly support the current description given in the materials and methods section.

An explanation indicating the reasoning of the choice for the 3 selected standard assays is missing.

Table 2: there are no numbers in italic.

Please check the use of a/an before an abbreviation.

A more extended explanation on why several samples are positive for multiple serotypes in the FRNT test is needed. Crossreactivity? infection with multiple serotypes?

In line 251 is is mentioned 'rather good'. This does not say much. Please specify what is meant with this.

In the first paragraph of the discussion an overall performance of the ViroTrack system is given. However, a discussion on whether this is sufficient for application in the field is missing. 

Reviewer 3 Report

This manuscript evaluates a commercial Dengue diagnostic test for recent and previous infections. The manuscript is well organized and the study is scientifically sound. The reviewer's major concern is the use of "biosensors" throughout the manuscript without proper definition. Biosensor is a very wide concept, which may also include the paper-based dengue serology RDTs and immunoassays used as references in the manuscript. In the Conclusion, the authors stated that "Biosensors-based dengue RDT such as ViroTrack has a huge potential of application". However, biosensors can be of vastly different natures. Potentially, not all biosensors would have the "huge potential" as the authors claimed. It is important that the authors clarify what "biosensors" refer to in the manuscript.

Round 2

Reviewer 1 Report

The revised version can be accepted.